# IPCP: Interpreter, Planner, Checker, and Painter Dialogue for Compositional Text-to-Image Generation

## Abstract

Text-to-image generation has advanced rapidly, but existing models still struggle with faithfully composing multiple objects and preserving their attributes in complex scenes. We propose **IPCP**, an interactive multi-agent dialogue framework with four specialized agents: Interpreter, Planner, Checker, and Painter that collaborate to improve compositional generation. The Interpreter adaptively decides between a direct text-to-image pathway and a layout-aware multi-agent process. In the layout-aware mode, it parses the prompt into attribute-rich object descriptors, ranks them by semantic salience, and groups objects with the same semantic priority level for joint generation. Guided by the Interpreter, the Planner adopts a divide-and-conquer strategy, incrementally proposing layouts for objects with the same semantic priority level while grounding decisions in the evolving visual context of the canvas. The Checker introduces an explicit error-correction mechanism by validating spatial consistency and attribute alignment, and refining the layouts before they are rendered. Finally, the Painter synthesizes the image step by step, incorporating newly planned objects into the canvas to provide richer context for subsequent iterations. Together, these agents address three key challenges: reducing layout complexity, grounding planning in visual context, and enabling explicit error correction. Extensive experiments on compositional benchmarks GenEval and DPG-Bench demonstrate that IPCP substantially improves text–image alignment, spatial accuracy, and attribute binding compared to existing methods.

## 1 Introduction

Text-to-Image (T2I) generation has emerged as a pivotal area in artificial intelligence, enabling the creation of visual content from textual descriptions (Rombach et al., 2022). However, current T2I models often struggle with user controllability, particularly concerning the reasonable arrangement and relationships of objects within the generated image (Zhang & Agrawala, 2023; Hertz et al., 2022). These models can exhibit numerical and spatial inaccuracies, leading to challenges in tasks such as faithful layout arrangement (Zheng et al., 2023) and maintaining compositional faithfulness, where the generated image accurately reflects the structure and relationships described in the text (Chefer et al., 2023; Liu et al., 2022).

To address the above limitations, existing works have explored the use of Large Language Models (LLMs) and agentic frameworks to assist in the generation of spatial layouts for images. Some approaches (Feng et al., 2023; Lian et al., 2023) leverage the planning capabilities of **LLMs** to interpret the input text and propose arrangements for the described objects, aiming to improve the structural coherence of the generated images. Furthermore, recent studies explore **agent-based frameworks** that uses multiple specialized LLM agents for text-to-image generation. For instance, MCCD (Li et al., 2025) demonstrates the effectiveness of multi-agent collaboration for compositional diffusion. Despite these advances, existing frameworks remain limited, often focusing only on text parsing, relying on a single agent, or reducing multi-agent setups to fixed pipelines, and consequently lack interaction, visual grounding, and autonomy.

However, planning layouts for scenes with multiple objects presents a significant challenge. First, global layout planning incurs quadratic relational complexity among objects, making it difficult

for a single planner to capture all dependencies (Zheng et al., 2023; Feng et al., 2023). Second, most approaches predict layouts without access to visual context, forcing the planner to "imagine" the scene in isolation, which often leads to incoherent or unrealistic arrangements. Third, most existing works use diffusion-based models and diffusion pipelines typically commit to a coarse global structure in early denoising steps, with fine details added only later (Hertz et al., 2022; Chefer et al., 2023). As a result, errors such as misplaced objects or incorrect attributes, once introduced early, are difficult to correct due to the lack of explicit error-correction mechanisms.

In this work, we propose an interactive multi-agent dialogue framework with four specialized agents: an Interpreter for generative mode selection and text decomposition, a Planner for incremental layout reasoning, a Checker for spatial and semantic verification, and a Painter for visual synthesis. The Interpreter determines whether to invoke the layout-free mode that connect the Painter directly or to activate the layout-aware mode for complex scenes. In the layout-free mode, the interpreter directly call a text-to-image painter for generation. On the contrary, in the layout-aware mode, unlike pipeline-based designs, our four agents engage in dynamic dialogue: it parses the text into attribute enriched object descriptions, ranks them by semantic salience, groups equal-priority objects for joint generation, and schedules iterative plans. Then the Planner incrementally proposes layouts one object (or group) at a time, the Checker validates spatial and semantic consistency against the text and evolving scene, and the Painter synthesizes the image step by step in a training-free and plug-and-play manner, with the evolving canvas providing crucial visual context for subsequent iterations.

Our framework effectively addresses the core challenges of prior methods. First, instead of performing global planning over all objects simultaneously, the Planner, guided by the Interpreter, adopts a divide-and-conquer strategy by reasoning about objects with the same semantic priority level at a time, thereby substantially reducing layout complexity. Second, the Planner leverages the evolving visual context from the Painter's canvas, ensuring that layout predictions are grounded in the actual scene rather than imagined in isolation. Third, the Checker introduces an explicit error-correction mechanism, validating object placement and attribute alignment, and applying the necessary adjustments to improve layout faithfulness. Together, these design choices reduce complexity, enhance robustness, and enable more faithful alignment between text and image.

We evaluate IPCP on the GenEval and DPG-Bench benchmarks. On GenEval, our framework sets a new state of the art with substantial improvements in compositional fidelity, object relations, and attribute binding over prior approaches. On DPG-Bench, which stresses long-context and multi-object reasoning, IPCP consistently outperforms recent baselines, achieving stronger spatial accuracy and text–image consistency. Qualitative comparisons further highlight these gains: whereas existing methods often misplace objects, miscount, or confuse attributes, IPCP generates coherent layouts and high-quality images closely aligned with the textual descriptions. These results demonstrate the effectiveness of our interactive multi-agent design for compositional image generation tasks.

In summary, our contributions are threefold:

- We introduce **IPCP**, an interactive multi-agent dialogue framework with four specialized agents: Interpreter, Planner, Checker, and Painter that collaborate dynamically instead of following a fixed pipeline.
- We propose three technical innovations: (i) a divide-and-conquer planning strategy that reduces layout complexity, (ii) grounding layout decisions in the evolving visual context for stronger spatial alignment, and (iii) an explicit error-correction mechanism via the Checker to enhance faithfulness.
- We achieve **state-of-the-art performance** on GenEval and DPG-Bench, showing clear gains in text–image consistency, spatial accuracy, and attribute binding.

## 2 RELATED WORK

**Text-to-Image Generation** The field of text-to-image (T2I) generation has seen rapid progress, initially driven by Vector Quantized GANs (VQGANs) (Esser et al., 2021) paired with CLIP guidance (Radford et al., 2021). The paradigm shifted significantly with the advent of diffusion models, which led to remarkable improvements in image quality and image-prompt alignment. Foundational models such as DALL-E (Ramesh et al., 2021), Imagen (Saharia et al., 2022), and Stable

Diffusion (Rombach et al., 2022) established the potential of this new approach. Subsequent efforts have focused on scaling, with models including (Betker et al., 2023; Podell et al., 2024; Labs, 2024; Duong & et al., 2025) further enhance performance. More recently, there is a growing trend of integrating Multimodal Large Language Models (MLLMs) directly into the generation process to improve prompt comprehension and contextual reasoning (Wang et al., 2024b; Wu et al., 2024a; Yang et al., 2024b; Hu et al., 2024b; Gani et al., 2023; Ding et al., 2021; Sun et al., 2023; Lee et al., 2024; Chen et al., 2025c; Qu et al., 2025; Lin et al., 2025a; Wu et al., 2025b; OpenAI, 2025; Gao et al., 2025; Wu et al., 2025a). However, these monolithic models still often struggle with fine-grained control over object composition and complex spatial relationships.

**Layout-to-Image Generation** To address the challenge of precise object placement, Layout-to-Image (L2I) generation conditions synthesis on explicit spatial information (Zheng et al., 2023; Feng et al., 2023; Zhou et al., 2024; Nuyts et al., 2024; Dahary et al., 2024; Jia et al., 2024; Ma et al., 2024; Lv et al., 2024; Zhang & et al., 2025), typically in the form of bounding boxes or segmentation masks. ControlNet (Zhang & Agrawala, 2023) and GLIGEN (Li et al., 2023) demonstrated spatial grounding in pre-trained diffusion models, while later works explored LLM-based layout generation (Lian et al., 2023; Feng et al., 2023), training-free constraints (Xie et al., 2023), and fine-grained regional controls (Cheng et al., 2024). These models require explicit spatial conditioning as an input condition, limiting their applicability when given only text input, unlike our method.

**Compositional Text-to-Image Generation** Ensuring compositional faithfulness, where the generated images reflect all objects, attributes, and relations in a prompt, remains a key challenge. Early works such as Composable Diffusion (Liu et al., 2022) and Attend-and-Excite (Chefer et al., 2023) combine concepts or refine attention guidance. Subsequent methods introduced layout reasoning as an intermediate step, for instance LayoutLLM-T2I (Qu et al., 2023), LLM Blueprint Gani et al. (2023), ALR-GAN (Tan et al., 2023), and LMD (Lian et al., 2023), which use LLMs or refinement modules to predict layouts that guide diffusion. RPG (Yang et al., 2024a) extends this idea by denoising subregions in parallel, while PlanGen (360CVGroup, 2024) integrates layout planning with synthesis. Most recently, GoT (Fang et al., 2025) employs a "Generation Chain-of-Thought" to produce a reasoning trace of semantic and spatial relations. While these approaches improve relational reasoning, they typically perform planning without visual feedback, making it difficult to resolve occlusion, depth, or other complex spatial interactions.

**Agent for Image Generation** Recent works have begun to explore agent-based paradigms for image generation, ranging from multi-agent prompt decomposition (Li et al., 2025), foreground-conditioned inpainting (Tianyidan et al., 2025), and self-correcting or interactive editing (Wang et al., 2024c; Wu et al., 2024b; Ma et al., 2025), to more recent directions such as self-improving agents (Wan et al., 2025), multicultural generation (Bhalerao et al., 2025), training-free pipelines (Chen et al., 2025a), and proactive multi-turn dialogue (Hahn et al., 2025). While these systems demonstrate the potential of agent designs, they are often limited by either fixed sequential pipelines (e.g., T2I-Copilot), planning solely from text without grounding (e.g., MCCD), or relying mainly on task scheduling or user queries without iterative visual feedback (e.g., Talk2Image, Proactive Agents). In contrast, our IPCP framework introduces a closed-loop multi-agent dialogue where the Planner, Checker, and Painter interact continuously with the evolving canvas, achieving stronger interactivity and more faithful compositional generation.

## 3 IPCP MULTI-AGENT SYSTEM

Given an input text $T$, our objective is to generate an image $I$ that faithfully aligns with the semantic content and spatial arrangement of the text. As shown in Figure 1, we propose IPCP, an interactive multi-agent dialogue framework in which four specialized agents: Interpreter, Planner, Checker, and Painter collaborate for compositional text-to-image generation. We will first introduce the details of agents collaboration and then each agent respectively in the remaining part of this section.

### 3.1 MULTI-AGENT COLLABORATION

To improve the generality of our approach, covering both general text-to-image cases without explicit layouts and more complex cases requiring layout planning, the Interpreter decides whether to enter the *layout-free* or the *layout-aware* mode. In the *layout-free* mode, the Interpreter directly

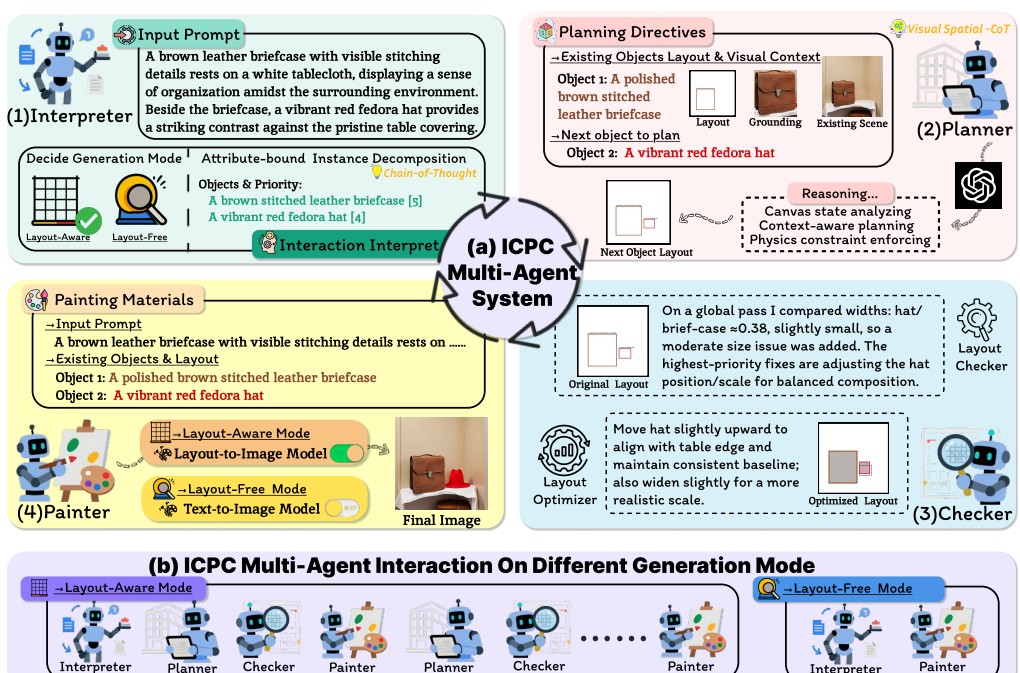

Figure 1: Overview of our proposed IPCP framework. (a) The multi-agent system consists of four specialized agents: (1) Interpreter, (2) Planner, (3) Checker, and (4) Painter. The Interpreter adaptively selects between layout-free and layout-aware modes; in the latter, the Planner incrementally proposes layouts, the Checker validates and refines them, and the Painter synthesizes the evolving canvas. (b) Illustration of the multi-agent interaction process in different generation modes, showing iterative collaboration in layout-aware mode and the direct pathway in layout-free mode.

invokes the Painter (a text-to-image model) to generate $I$ that aligns with $T$. In the *layout-aware* mode, the Interpreter first parses $T$ into attribute-rich object descriptors, ranks them by semantic importance, and groups objects of similar priority for joint generation. The generation then proceeds through a Planner–Checker–Painter loop, iterating once for each semantic priority level. In the $i^{th}$ iteration, Planner incrementally proposes layouts $L_i$ for the set of highest-salience objects at a time based on the existing objects grounding and scene visual context rather than the entire scene. The Checker then leverages both the text and the visual context to validate spatial consistency and objects semantic alignment, followed by layout refinement. The Painter synthesizes the image step by step, incorporating each newly planned object into the evolving canvas, which in turn provides essential visual context for subsequent iterations. After $N$ layout iterations, the final image $I$ is produced, closely aligned with the input text $T$.

In the IPCP layout-aware mode, the Planner adopts a divide-and-conquer strategy by reasoning about objects of the same semantic priority level at a time, which reduces layout complexity. Guided by the evolving canvas from the Painter, it grounds layout predictions in the actual scene rather than imagining them in isolation. The Checker further validates object placement and attribute alignment and refines the layout. This collaborative loop alleviates the burden of spatial planning and yields images that more faithfully reflect the input text, particularly in complex object arrangements.

## 3.2 INTERPRETER, PLANNER, CHECKER, AND PAINTER

In this section, we detail the four specialized agents: interpreter, planner, checker, and painter, each of whom is responsible for a distinct role in our generation process.

**Interpreter** To accurately represent complex scenes as a structured input for our iterative framework, we introduce an Interpreter agent to process $T$ for the agent system. The Interpreter first infers the relative importance of objects from $T$, and then decides whether to invoke the Painter directly for detail fidelity or to activate the multi-agent dialogue for layout-precise generation.

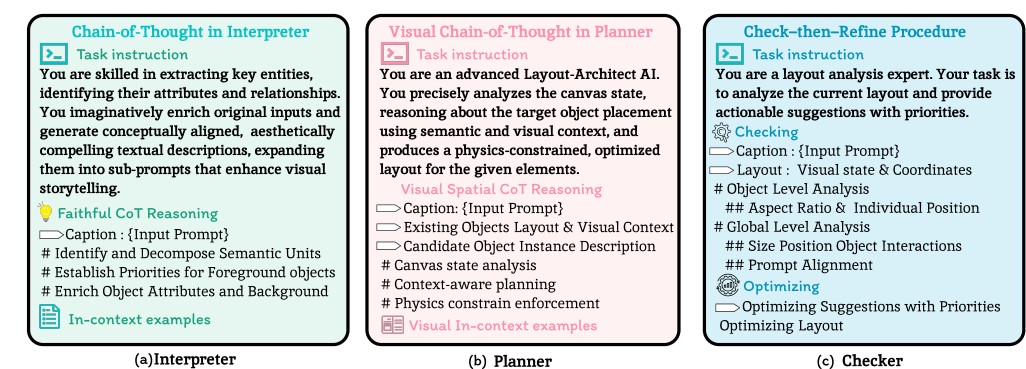

Figure 2: Prompting structure for agents in our framework.

As illustrated in Figure 1(a), in the layout-aware mode the Interpreter decomposes the text prompt $T$ into structured, semantically rich object descriptors and prepares them for downstream planning. Specifically, we leverage large language models (LLMs) with chain-of-thought (CoT) prompting, guided by task instructions as shown in Figure 2(a). The process follows three steps: (i) **Identify and decompose** the prompt into distinct semantic units; (ii) **Establish priorities** by ranking objects according to their semantic salience and grouping items with the same semantic-level for joint generation; and (iii) **Enrich attributes and background** through CoT-guided queries, yielding detailed descriptors of objects and their relations. The Interpreter then assigns the highest-priority objects of the current round to the iteration, enabling interactive generation with the Planner, Checker, and Painter in the generation loop.

**Planner**  The workflow of Planner is shown in Figure 1 (b). At $i^{th}$ iteration, Planner aims to plan the layout $L_i$ of the objects at the $i^{th}$ priority ranked by the Interpreter. Motivated by the multimodal chain-of-thought (Zhang et al., 2024), we propose a stepwise visualization chain-of-thought (VCoT) displayed in Figure 2 (b) for layout planning. We employ GPT-5 as our MLLM for VCoT.

VCoT takes as input the global text prompt $T$, the description of the $i^{th}$ priority objects, the layouts generated in the previous $i-1$ iterations, and the partial image $I_{i-1}$ rendered by Painter. It also incorporates object grounding, establishing correspondences between textual entities and image regions in $I_{i-1}$, which mitigates the inherent insensitivity of LLMs to spatial coordinates (You et al., 2024) and enables reliable object localization.

We formulate our CoT reasoning as three steps: Canvas state analysis, Context-aware planning, and Physics constrain enforcement. In the "Canvas state analysis" stage, guided by the rich visual context of objects grounding, the image $I_{i-1}$ and other inputs, Planner meticulously analyzes the spatial layout of existing objects to gain a comprehensive visual understanding the current state of the scene. Afterwards, in the "Context-aware planning" stage, based on the existing canvas state, the MLLM planner leverages its embedded world knowledge to reason about the plausible interactions between the candidate object $O_i$ and the existing scene composition $(O_0, \ldots, O_{i-1})$. Further, to maintain physical plausibility and scene coherence, Planner incorporate a "Physics constrain enforcement" module to prompt the MLLM to take physical and contextual constraints into account, which encourages realistic object placement to reflect real-world interactions and prevents issues like floating objects or improbable contacts. Please refer to Appendix A for more details on VCoT.

**Checker**  At each iteration $i$, the Checker performs a two-stage check–then–refine procedure illustrated in Figure 1(c). **In the first stage**, it analyzes the current proposal $L_i$ and conducts checking at two levels: object level and global level; as shown in Figure 2 (c). At the object level, it inspects size, scale, and boundary coverage, while at the global level it evaluates relative placement, inter-object relations, and spatial plausibility. Based on these assessments, the Checker updates $L_i$ accordingly. **In the second stage**, it reviews all layouts from the previous iterations $\{L_1, \ldots, L_i\}$ to identify cross-object conflicts such as overlaps, occlusion ordering, or scale drift, and makes corresponding adjustments. The refined layout is then passed to the Painter for rendering.

**Painter**  In IPCP, the Painter supports two modes. In layout-free mode, it invokes a text-to-image (T2I) model to synthesize the image $I$ directly from the prompt. In layout-aware mode, it uses a

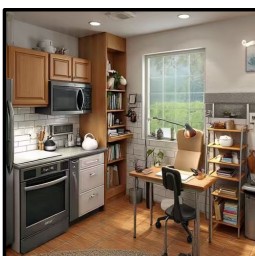 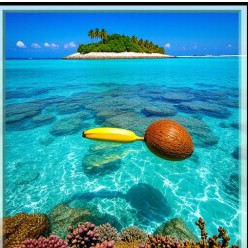 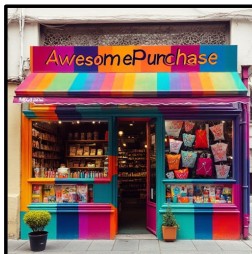 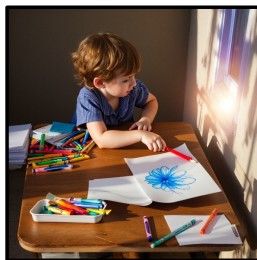

View of kitchen and study area, kitchen with tiled countertop and floor, oven and overhead microwave, spice rack, kettle, and in study area, wood floor, shelves, window, desk with desk lamp, and a padded, rolling office chair.

In a clear blue tropical sea, a ripe yellow banana bobs on the gentle waves alongside a brown, hairy coconut. The fruit duo is surrounded by vibrant coral visible beneath the water's surface. Near the horizon, one can spot a small island with lush green palm trees swaying in the breeze.

a brightly colored storefront with large, bold letters spelling out 'AwesomePurchase' above the entrance. The shop's window displays are neatly arranged with an array of products, and a small, potted plant sits to the left of the door. The facade of the building is a clean, modern white, contrasting with the vibrant signage.

A young child with brown hair, focused intently, sits near a wooden table scattered with colorful crayons and paper. In their small hand is a bright red pencil, with which they are diligently drawing a vibrant blue flower that's taking shape on the white sheet before them. Sunlight filters through a nearby window, casting a warm glow on the child's artwork.

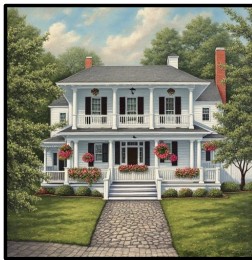 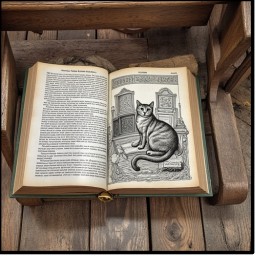 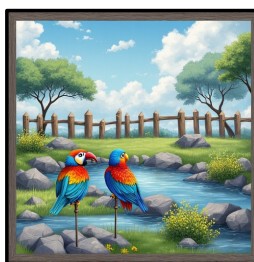 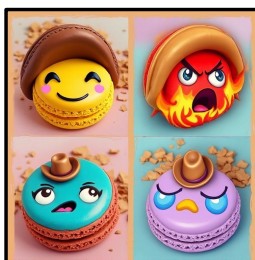

A picturesque painting depicting a charming white country home with a spacious wrap-around porch adorned with hanging flower baskets. The house is set against a backdrop of lush greenery, with a cobblestone pathway leading to its welcoming front steps. The porch railing is intricately designed, and the home's windows boast traditional shutters, adding to the aesthetic of the scene.

A spacious, open book lies flat on a wooden table, its pages filled with blocks of text and a large, detailed illustration of a cat on the right side. The illustration depicts a gray feline with intricate patterns, lounging amidst a backdrop of sketched furniture. The left page is densely packed with small, black font, narrating a story that accompanies the image, and the edges of the book's pages show signs of frequent use.

In the foreground, two birds with vibrant feathers are perched upon rugged grey rocks that jut out near a tranquil pond with lush green plants at the water's edge. In the midground, a rustic wooden fence creates a boundary line, subtly dividing the natural scene from the world beyond. The background extends into a vast expanse of soft blue sky dotted with tufts of white clouds, stretching far into the horizon.

A playful collection of 2x2 emoji icons, each resembling a vibrant macaron with a distinct facial expression. The top left macaron is a sunny yellow with a beaming smile, while the top right is a fiery red with furrowed brows and an angry scowl. Below them, the bottom left is a bright blue with wide, surprised eyes, and the bottom right is a soft lavender with a tearful, sobbing face. Each of the macaron emojis is whimsically topped with a miniature brown cowboy hat, adding a touch of whimsy to their appearance.

Figure 3: Generative results of our IPCP framework.

layout-to-image (L2I) model conditioned on the current layout. Across iterations $i$, the Painter incrementally renders the canvas by integrating each newly confirmed object, providing visual context for subsequent steps. At the final iteration, the Painter renders the final image $I$.

The models used by Painter is designed to be plug-and-play, allowing any text-to-image(T2I) and layout-to-image (L2I) model to be seamlessly integrated without additional training. In this paper, we use Flux Labs (2024) for T2I model and 3DIS Zhou et al. (2024) for L2I model. Note that our model is designed to be compatible with other, potentially more advanced L2I models, which could further improve our text-to-image generation performance.

## 4 EXPERIMENTS

### 4.1 DATASET AND METRICS

We rigorously evaluated our IPCP framework using two benchmark datasets: GenEval (Ghosh et al., 2023b) and DPG-Bench (Hu et al., 2024a). GenEval, a standard for assessing text-to-image generation quality and text-image alignment, provides six metrics including object presence, attribute binding, counting, and spatial relationships. We report the overall GenEval Score, along with its sub-scores, to quantify our model's performance in each of these aspects. For DPG-Bench, which is designed to evaluate a model's ability to follow lengthy and dense prompts describing multiple objects with diverse attributes and relationships, we follow its established protocol, using an MLLM to adjudicate the generated images based on a series of questions. We report the overall DPG-Bench score, which is the average score across all prompts, along with scores for its main sub-categories.

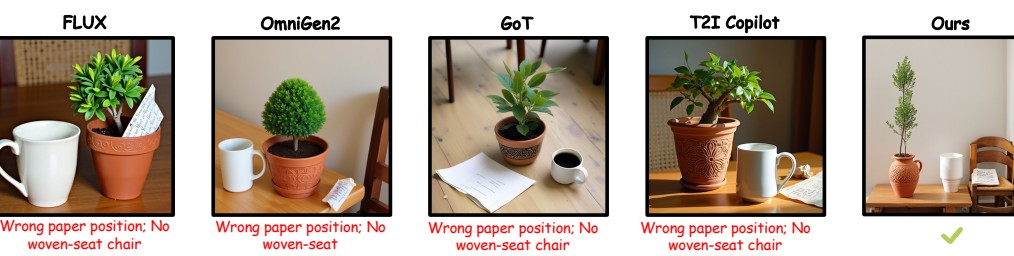

A small, vibrant green tree sits snugly within a terracotta pot that features intricate patterns etched into its surface. The pot is placed to the left of a simple white ceramic cup with a delicate handle, both resting on a wooden countertop. To the side of the cup is a chair with a woven seat, and the tree in the pot shares this proximity with the chair as well. Perched precariously on the edge of the chair is a crumpled piece of paper, the handwriting upon it partially visible, creating a tableau of everyday items in close association.

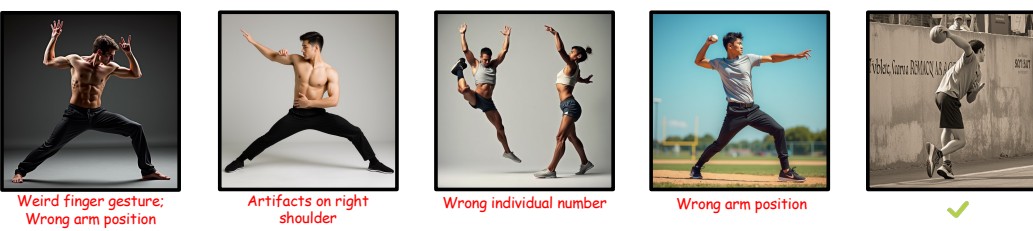

A dynamic stance captured in a moment of intense action shows an individual with their legs spread apart for balance. Their right arm is drawn back, poised in a throwing position, with their hand just below the level of their head, ready to launch. The left arm is relaxed and lowered, the elbow bent, and the hand gently resting on the stomach area, creating a counterbalance to the tension in the right arm.

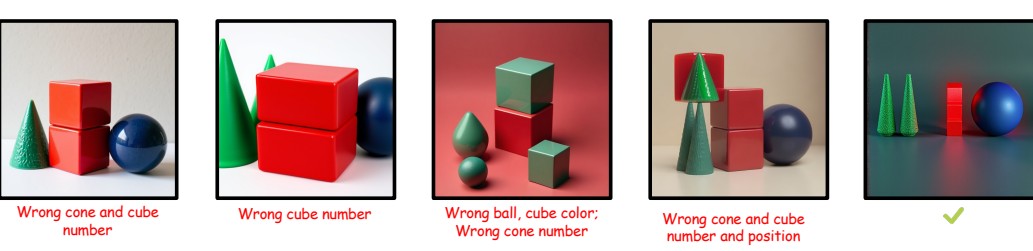

In the center of the composition, there is a neatly arranged stack of three vibrant red cubes, each with a smooth, glossy finish that reflects the ambient light. To the right of this stack, there is a deep blue sphere with a matte texture, providing a stark contrast to the geometric sharpness of the cubes. On the left side, two emerald green cones with a slightly textured surface are positioned, their pointed tips directed upwards, creating a symmetrical balance in the arrangement.

Figure 4: Qualitative comparison with existing methods.

## 4.2 Our Results and Comparisons with Existing Methods

As shown in Figure 3, our IPCP framework is capable of handling a wide range of challenging scenarios, including multi-object compositions, diverse visual styles, highly complex layouts with object interactions, and long descriptive text prompts. The generated images remain highly consistent with the input descriptions while maintaining high visual quality. Additional results on DPG-Bench and GenEval are provided in the supplementary material. We also present our generative results on COCO dataset Lin et al. (2014) and additional customized prompts to demonstrate the framework's ability to generalize to complex scenes and inter-object relationships in the supplementary material.

For quantitative evaluation, we compare IPCP with a broad set of recent state-of-the-art models on GenEval and DPG-Bench (Tables 1 and 2). The baseline models encompass a broad range of state-of-the-art text-to-image generation approaches, including representative T2I models such as DALL-E 3 (Shi et al., 2020) and FLUX (Labs, 2024), recent multimodal large models like GPT Image 1 [High] (OpenAI, 2025) and OmniGen2 (Wu et al., 2025b), layout-aware methods such as GoT (Fang et al., 2025) that perform explicit bounding box reasoning, and multi-agent frameworks like T2I-Copilot (Chen et al., 2025a). These models collectively represent diverse paradigms in the field, from direct generation to structured reasoning and interactive agent collaboration. IPCP achieves the best overall results, demonstrating the effectiveness of our interactive multi-agent design.

For qualitative comparisons, we further compare IPCP with representative methods from three categories: general text-to-image generation FLUX (Labs, 2024), vision-language models Omni-Gen2 (Wu et al., 2025b), one of the most recent open-source compositional image generation works that explicitly reasons bounding boxes GoT (Fang et al., 2025), and the most recent multi-agent

Table 1: Performance comparison on the GenEval (Ghosh et al., 2023a). Best results are marked in **bold**. Column names are abbreviated to fit the page.

| Model | Single Obj. | Two Obj. | Counting | Colors | Position | Color Attri. | Overall↑ |
|---|---|---|---|---|---|---|---|
| PixArt-Σ (Chen et al., 2024) | 0.98 | 0.50 | 0.44 | 0.80 | 0.08 | 0.07 | 0.48 |
| Emu3-Gen (Wang et al., 2024a) | 0.98 | 0.71 | 0.34 | 0.81 | 0.17 | 0.21 | 0.54 |
| SDXL (Podell et al., 2023) | 0.98 | 0.74 | 0.39 | 0.85 | 0.15 | 0.23 | 0.55 |
| GoT Fang et al. (2025) | 0.99 | 0.69 | 0.67 | 0.85 | 0.34 | 0.27 | 0.64 |
| DALL-E 3 (Shi et al., 2020) | 0.96 | 0.87 | 0.47 | 0.83 | 0.43 | 0.45 | 0.67 |
| FLUX.1-dev (Labs, 2024) | 0.99 | 0.81 | 0.79 | 0.74 | 0.20 | 0.47 | 0.67 |
| Janus-Pro-1B (Chen et al., 2025c) | 0.98 | 0.82 | 0.51 | 0.89 | 0.65 | 0.56 | 0.73 |
| SD3-Medium (Esser et al., 2024) | 0.99 | 0.94 | 0.72 | 0.89 | 0.33 | 0.60 | 0.74 |
| TokenFlow-XL (Qu et al., 2025) | 0.95 | 0.60 | 0.41 | 0.81 | 0.16 | 0.24 | 0.55 |
| UniWorld-V1 (Lin et al., 2025a) | 0.99 | 0.93 | 0.79 | 0.89 | 0.49 | 0.70 | 0.80 |
| GPT Image 1 [High] (OpenAI, 2025) | 0.99 | 0.92 | 0.85 | 0.92 | 0.75 | 0.61 | 0.84 |
| **IPCP(Ours)** | **1.00** | **0.96** | **0.94** | **0.97** | **0.95** | **0.81** | **0.94** |

Table 2: Performance comparison on the DPG-Bench (Hu et al., 2024a). Best results are in **bold**.

| Model | Global | Entity | Attribute | Relation | Other | Overall↑ |
|---|---|---|---|---|---|---|
| Hunyuan-DiT (Li et al., 2024) | 84.59 | 80.59 | 88.01 | 74.36 | 86.41 | 78.87 |
| PixArt-Σ (Chen et al., 2024) | 86.89 | 82.89 | 88.94 | 86.59 | 87.68 | 80.54 |
| DALL-E 3 (Shi et al., 2020) | 90.97 | 89.61 | 88.39 | 90.58 | 89.83 | 83.50 |
| SD3-Medium (Esser et al., 2024) | 87.90 | 91.01 | 88.83 | 80.70 | 88.68 | 84.08 |
| FLUX.1-dev (Labs, 2024) | 74.35 | 90.00 | 88.96 | 90.87 | 88.33 | 83.84 |
| GoT (Fang et al., 2025) | 83.58 | 82.16 | 80.07 | 87.81 | 65.25 | 73.53 |
| TokenFlow-XL (Qu et al., 2025) | 78.72 | 79.22 | 81.29 | 85.22 | 71.20 | 73.38 |
| T2I-Copilot (Chen et al., 2025a) | 87.50 | 81.74 | 81.07 | 86.94 | 48.28 | 74.34 |
| Emu3-Gen (Wang et al., 2024a) | 85.21 | 86.68 | 86.84 | 90.22 | 83.15 | 80.60 |
| UniWorld-V1 (Lin et al., 2025b) | 83.64 | 88.39 | 88.44 | 89.27 | 87.22 | 81.38 |
| BLIP3-o 8B (Chen et al., 2025b) | - | - | - | - | - | 81.60 |
| OmniGen2 (Wu et al., 2025b) | 88.81 | 88.83 | 90.18 | 89.37 | 90.27 | 83.57 |
| **IPCP(Ours)** | 84.78 | 90.15 | 87.55 | 92.92 | 84.38 | **85.17** |

text-to-image generation framework T2I-Copilot (Chen et al., 2025a). As shown in Figure 4, existing approaches often suffer from misplaced objects, incorrect counts, or attribute artifacts, while our method produces coherent layouts and faithful compositions closely aligned with the textual descriptions.

### 4.3 EFFICIENCY OF IPCP

We report agent-usage statistics on DPG-Bench with 1,074 images. As shown in Table 3, the Interpreter, Planner, Checker, and Painter are each invoked only a few times per generation (1.00, 1.52, 1.62, and 1.95 on average, respectively), which is far fewer than the average number of objects present in a scene (2.79). This efficiency arises because the Interpreter groups objects of the same semantic level, enabling multiple objects to be processed within a single round. This design substantially improves efficiency while maintaining strong performance.

### 4.4 ABLATION STUDIES

We conduct ablation experiments on DPG-Bench to evaluate the contribution of each component in our framework. As shown in Table 4 and Figure 5, starting from the layout-free baseline, introducing layout-aware planning allows the LLM to explicitly generate layout plans and adopt a divide-and-conquer strategy over multiple objects rather than attempting global planning at once, thereby reducing reasoning complexity. Adding visual context enables the Planner to leverage the partially generated scene as grounding when placing the next set of semantically prioritized objects, which enhances spatial coherence. The Checker provides explicit error correction by detecting misplacements and attribute mismatches, further improving entity and attribute faithfulness. Our full

Table 3: The average number of agent calls and objects in each DPG-bench image.

| Agent | Interpreter | Planner | Checker | Painter | Number of Objects |
|---|---|---|---|---|---|
| Avg. calls / generation | 1.00 | 1.52 | 1.62 | 1.95 | 2.79 |

Table 4: Quantitative ablation study.

| Model | Global | Entity | Attribute | Relation | Other | Overall↑ |
|---|---|---|---|---|---|---|
| Layout-free mode | 84.50 | 84.44 | 86.15 | 90.87 | 75.60 | 77.60 |
| + Layout-aware mode | 79.94 | 89.32 | 87.27 | 92.37 | 80.65 | 82.61 |
| + Visual context | 88.89 | 88.72 | 89.32 | 95.95 | 66.67 | 84.51 |
| + Checker (IPCP) | 84.78 | 90.15 | 87.55 | 92.92 | 84.38 | **85.17** |

**Layout Free**  **+ Layout Aware**  **+ Visual context**  **+ Checker (Ours)**

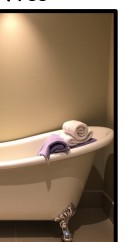 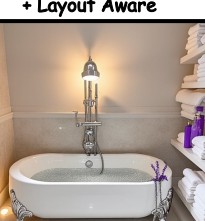 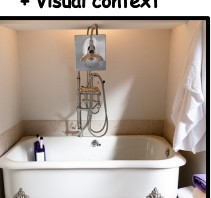 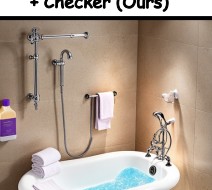

No lavender-scented bath products, water; Wrong showerhead position     Weird mixture of light and showerhead     Weird mixture of light and showerhead     ✓

Under the warm glow of an overhead light, a shiny chrome showerhead is poised above a pristine white bathtub with clawed feet. The porcelain surface of the tub is speckled with droplets of water, ready to embrace the evening's tranquility. To the side of the bathtub, an assortment of lavender-scented bath products and fluffy towels are neatly arranged, hinting at the luxurious bath time ritual that awaits.

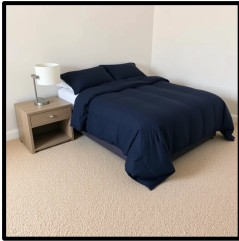 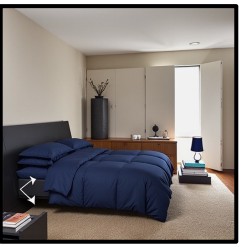 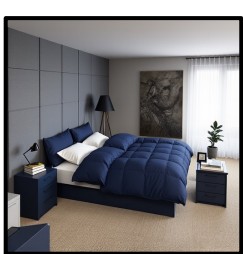 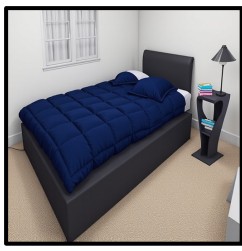

No books     Weird mixture of window and closet; Wrong lamp position     Weird nightstand position     ✓

Adjacent to each other in a room, a large rectangular bed draped in a navy-blue comforter sits parallel to a square-shaped nightstand with a matte finish. The nightstand holds an angular lamp and a small stack of hardcover books. The two pieces of furniture are positioned on a plush beige carpet that covers most of the floor space.

Figure 5: Qualitative ablation results.

IPCP model achieves the best overall balance, producing visually coherent scenes with stronger alignment between objects, attributes, and relations.

## 5 CONCLUSION

In this paper, we introduced IPCP, an interactive multi-agent dialogue framework for compositional text-to-image generation. IPCP brings together four specialized agents: the Interpreter that decomposes the text prompt, the Planner that reasons about object layouts, the Checker that verifies spatial and semantic consistency, and the Painter that renders the final image. By coordinating these roles, IPCP addresses key challenges in complex scene generation, including layout reasoning, grounding in evolving visual context, and explicit error correction. Evaluations on GenEval and DPG-Bench show that IPCP achieves state-of-the-art performance, with notable gains in text–image consistency, spatial accuracy, and attribute binding compared with existing approaches. Limitations are discussed in the Supplementary Material.

ETHICS STATEMENT

Our work focuses on developing an interactive multi-agent framework for compositional text-to-image generation. While the method demonstrates strong ability to generate high-quality, realistic images, we recognize that such capabilities may raise potential ethical concerns. In particular, highly realistic image generation could be misused for creating misleading or harmful visual content. To mitigate these risks, our experiments are conducted only on publicly available datasets (e.g., GenEval, DPG-Bench), and all results are intended purely for academic research. We do not generate or release images involving sensitive personal data, political topics, or harmful scenarios.

REPRODUCIBILITY STATEMENT

We provide detailed descriptions of our proposed frameworks in Section 3, including all architectural components, agent interactions, and the full pipeline for generating images from text inputs. Section 4 details the experimental setup, including datasets and experimental configurations. **Codes will be released upon publication.**

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

## A    DETAILS OF VISUAL CoT (VCoT) REASONING

Figure 6 illustrates the detailed reasoning steps of our Visual Spatial CoT (VCoT) module. Given the current canvas state and instruction, VCoT performs the following steps:

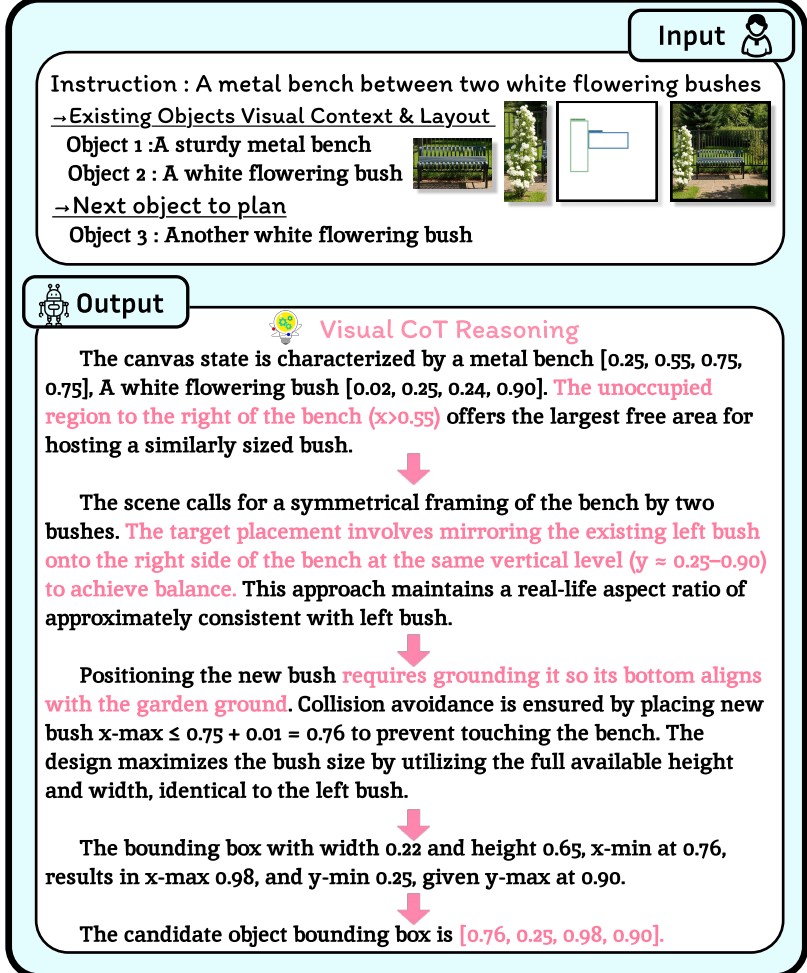

Figure 6: A reasoning example of the VS-COT module in our framework

1. **Scene parsing.** The module first parses the existing layout, identifying objects and their bounding boxes. For example, a metal bench is located at [0.25, 0.55, 0.75, 0.75], and a white flowering bush is positioned at [0.02, 0.25, 0.34, 0.90].

2. **Free-space identification.** VCoT then analyzes the unoccupied regions of the canvas. In this case, the right side of the bench (x ≈ 0.25–0.95) is identified as the feasible area for placing another bush.

3. **Symmetry reasoning.** To maintain balanced composition, the target placement is chosen to mirror the existing bush on the left, aligning along the same vertical level (y ≈ 0.90).

4. **Grounding and constraint enforcement.** The candidate bounding box is grounded to the garden floor (bottom aligned), ensuring physical plausibility. Collision constraints are checked to avoid overlap with the bench by setting the bush's x-max ≤ 0.76.

5. **Bounding box refinement.** The box dimensions are adjusted to maximize use of available space while keeping consistency with the left bush. The final bounding box is given as [0.76, 0.25, 0.98, 0.90].

Through these step-by-step spatial reasoning processes, VCoT generates placements that are physically valid, compositionally balanced, and text-aligned.

## B  ADDITIONAL RESULTS ON DPG BENCH

Figure 7, 8, 9 and 10 presents six representative cases from DPG-Bench, covering diverse everyday and imaginative scenarios. The prompts feature multiple objects, long textual descriptions, and complex interactions, such as indoor arrangements (books, couches, kitchen scenes), dynamic activities (a surfer riding waves, a man and dog playing with a frisbee), and even global landmarks (Sydney Opera House, Eiffel Tower, Mount Everest). Across all these cases, IPCP produces images that remain highly faithful to the text, accurately capturing object positions, counts, and attributes while preserving visual coherence and style. These results further confirm that our multi-agent framework scales effectively to the challenging long-context, multi-object compositions posed by DPG-Bench.

Notably, the inclusion of layout-free examples, where instance decomposition and bounding boxes are not available, demonstrates the flexibility of our approach, which can adaptively select between layout-free and layout-aware modes to accommodate varying text prompt requirements.

## C  ADDITIONAL RESULTS ON GENEVAL

Figures 11 and 12 present our generative results on GenEval. IPCP accurately captures object attributes and spatial relationships across diverse scenarios. The framework successfully generates realistic and coherent images, such as a dog positioned to the right of a teddy bear, as well as indoor scenes involving a brown dining table and a white sofa frame.

Notably, IPCP exhibits strong generalization capabilities in handling unconventional or semantically implausible compositions, such as a majestic brown horse placed alongside a leather couch or a computer keyboard, which are rare or unrealistic in real-world contexts. These cases highlight the model's robustness in following uncommon or imaginative prompts.

## D  ADDITIONAL RESULTS ON COCO

Figures 13, 14, 15, 16 present additional qualitative examples from COCO Lin et al. (2014). The cases highlight key challenges of compositional generation, including precise object counts (e.g., multiple sinks, zebras, and giraffes), correct spatial relations (e.g., airplanes with dogs, people watching jets), and faithful attribute binding (e.g., bathroom themes, kitchen appliances). In all these scenarios, IPCP produces images that remain consistent with the descriptions, confirming its strong capability in handling fine-grained compositional reasoning on real-world scenes.

## E  ADDITIONAL RESULTS ON CUSTOMIZED PROMPTS INVOLVING OBJECT INTERACTIONS

Our method demonstrates strong capability in generating images with highly interactive object pairs, where precise spatial alignment and semantic coherence are critical. As shown in Figure 17, IPCP successfully handles complex relational prompts such as "a man sit in a car", "two person hugging", and "a person driving from a mug sitting at a table". In each case, the generated images exhibit accurate object positioning and natural interactions—such as the person being properly seated inside the car, the hugging figures sharing appropriate body contact and pose alignment, and the person interacting with the mug in a plausible tabletop setting. These results highlight IPCP's ability to model strong inter-object dependencies and contextual relationships.

## F  LIMITATION

While IPCP Dialogue demonstrates significant progress in compositional text-to-image generation, it still has several limitations:

First, the multi-agent system, while beneficial for quality and compositional accuracy, introduces a computational overhead. The framework requires more processing time compared to single-pass methods due to the multi-agent calls. However, empirical analysis 4.3 shows IPCP Dialogue still achieves competitive inference efficiency, outperforming many existing methods despite its iterative nature. Further optimization of the multi-agent loop remains a key area for future work.

Second, the performance of our Painter is inherently dependent on the underlying T2I and L2I models. This dependency means that limitations of the base models, such as imperfect attribute rendering or biased visual priors, may propagate into IPCP, e.g., "a radish with black skin". Conversely, it also indicates that IPCP will naturally benefit from future advances in text-to-image and layout-to-image generation.

Third, the Planner and Checker rely on multimodal LLMs for layout reasoning and error detection, making the system susceptible to LLM-specific issues such as hallucination and overconfidence in incorrect layouts. These limitations may lead to invalid object placements or missed corrections, especially in highly compositional or ambiguous prompts. Conversely, it also indicates that IPCP will naturally benefit from future advances in more reliable and grounded LLMs with reduced hallucination tendencies.

Finally, as with most iterative frameworks, IPCP may be affected by error accumulation across iterations. For instance, small placement inaccuracies in early steps can propagate if not fully corrected by the Checker. Nevertheless, our design explicitly mitigates this risk by introducing verification and refinement mechanism, and we observe that the overall error accumulation is significantly lower than in single-pass generation pipelines, as shown in Tables 1 and 2.

## G  THE USE OF LARGE LANGUAGE MODELS (LLMS)

To enhance the clarity and readability of this manuscript for global audiences, we employed Large Language Models (LLMs) as a tool for language refinement. The models were used for grammatical correction, stylistic improvements, and rephrasing complex sentences to ensure our technical contributions were communicated as clearly as possible. The authors maintained full editorial control throughout this process. The intellectual content, including all research ideas, methodologies, and conclusions, remains the exclusive work of the authors, who bear full responsibility for the final manuscript.

A powerful black red steam locomotive
A billowing white steam plume
A long desert railway track
Several small sand clouds

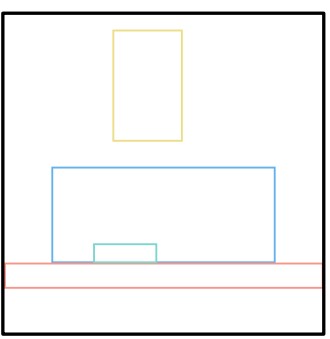 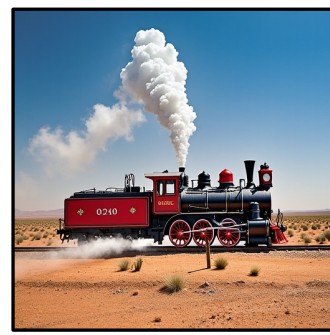

A powerful steam locomotive with a black and red exterior, billowing white steam as it speeds along the tracks through a vast, sandy desert landscape. The locomotive's wheels kick up small clouds of sand, and the clear blue sky stretches endlessly above. No other vehicles or structures are in sight, just the occasional cactus dotting the horizon.

Layout-free mode : instance decomposition not available

layout-free mode : bounding boxes not available

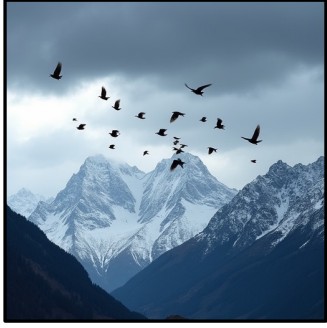

In the distance, towering black mountains with their peaks blanketed in thick layers of snow stand majestically. Against this dramatic backdrop, a flock of black birds is captured in their dynamic mid-flight, crisscrossing the scene with elegance and energy. Above them, the sky is a tapestry of deep grays clashing with the remnants of serene blue, creating a striking contrast that defines the horizon.

Several colorful vanilla-specked ice cream scoops
A clear glass bowl
A warmly lit microwave interior

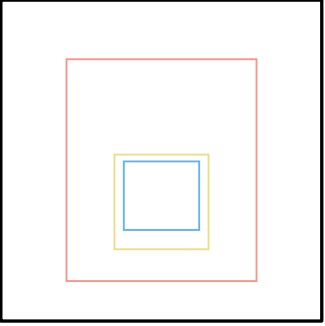 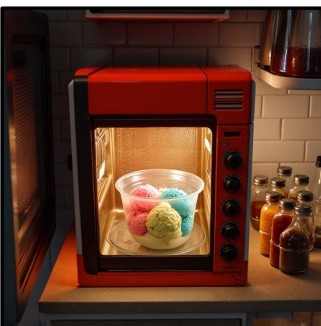

Inside the microwave sits a clear glass bowl, filled to the brim with scoops of colorful ice cream with visible flecks of vanilla beans. The microwave's interior light casts a warm glow on the ice cream, which threatens to melt if the door were to remain closed for long. It's an odd place for a cold dessert that's usually served at a chilly temperature to avoid its creamy contents from turning into a soupy mess. The microwave is positioned on a countertop, surrounded by assorted kitchen gadgets and a spice rack full of various seasonings.

Figure 7: Generative results of our IPCP on DPG Bench.

| Instance descriptions | Layout | Image |
|---|---|---|

A balancing surfer
A bright yellow surfboard
A cresting ocean wave
A reflective glass skyscraper
A beige brick skyscraper

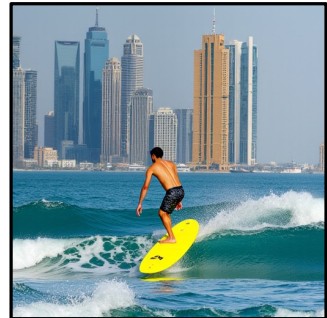

an individual balancing on a bright yellow surfboard, riding the crest of an ocean wave. parallel to the shore, a series of tall buildings stand in close proximity to one another, creating a dense urban skyline. the closest building has a reflective glass facade, while the one alongside it features beige brickwork.

A dull-gleaming cast-iron kettle
A floral ceramic teapot
A rough-hewn wooden table
A woven dried-flower basket
A partially-drawn curtained window

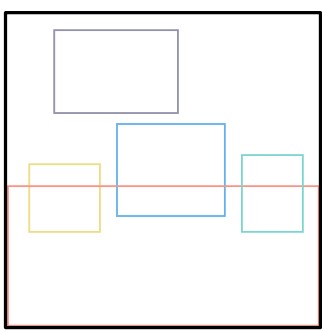 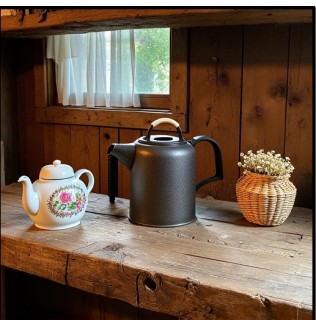

An old-fashioned kitchen setting with a cast-iron kettle and a ceramic teapot sitting atop a rough-hewn, wooden table that bears the marks and patina of age. The kettle's metallic surface has a dull gleam, reflecting the warm ambient light, while the teapot, adorned with a floral pattern, adds a touch of nostalgia to the setting. In the background, there is a window with curtains partially drawn, allowing for a soft natural light to fill the room. Nearby, a woven basket filled with dried flowers accentuates the rustic charm of the cozy interior.

Layout-free mode : instance decomposition not available

layout-free mode : bounding boxes not available

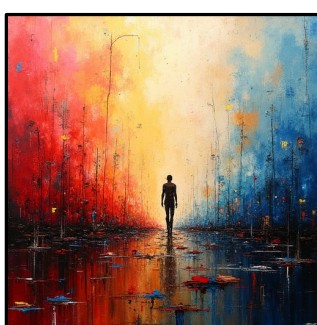

An abstract oil painting that depicts a chaotic blend of vibrant colors and swirling patterns, giving the impression of a vast, disorienting landscape. The canvas is filled with bold strokes of reds, blues, and yellows that seem to clash and compete for space, symbolizing the complexity and confusion of navigating through life. Amidst the turmoil, a small, indistinct figure appears to be wandering, searching for direction in the overwhelming expanse.

Figure 8: Generative results of our IPCP on DPG Bench.

| Instance descriptions | Layout | Image |
|---|---|---|

A smooth patterned ceramic vase
A dense leafy bush
A wooden garden bench
A paved street

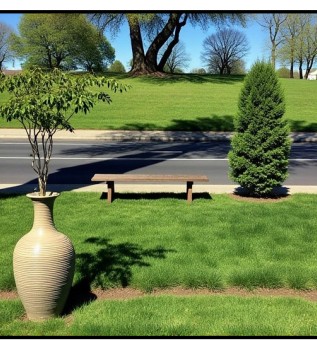

A picturesque outdoor scene featuring a ceramic vase prominently placed to the left of a lush, green lawn. The vase, with its smooth texture and intricate patterns, stands in the foreground, with the expansive, clear blue sky stretching overhead. Beyond the vase, a wooden bench can be seen, slightly obscured by the vase's presence. To the right, a dense, leafy bush rises up against the sky, situated just above a paved street that runs adjacent to the bush.

A barefoot man in striped shorts
A playful black and white dog
A spinning white frisbee midair

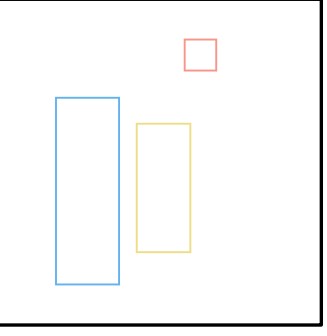
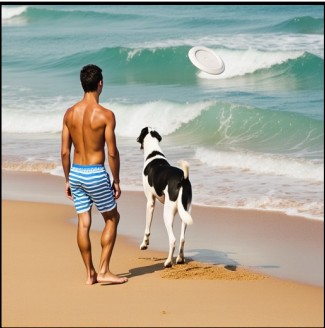

A beach scene captures a man, clad in blue and white striped swim shorts, standing barefoot on the warm, golden sand. To his side, a playful black and white dog, with its gaze fixed on an object in the sky, waits in anticipation. Suspended in the air above them is a spinning white frisbee, creating a dynamic moment of play and excitement just off the coast, where the gentle waves lap at the shore.

A white-shelled Sydney Opera House
An intricate iron Eiffel Tower
A snow-capped Mount Everest

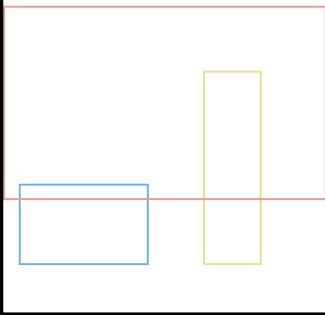
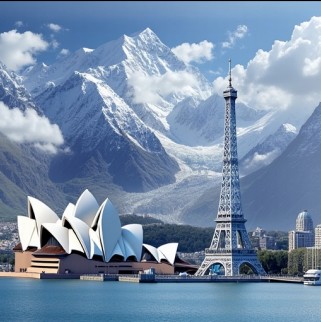

An imaginative scene where the iconic Sydney Opera House, with its white sail-like shells, sits prominently on the left. To the right, the Eiffel Tower, constructed of intricate iron lattice work, towers over the landscape. Behind both landmarks, the majestic Mount Everest looms, its snow-capped peak piercing the sky.

Figure 9: Generative results of our IPCP on DPG Bench.

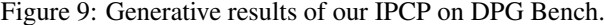

A green-pink floral spring cartoon calendar
A sunny blue summer cartoon calendar
An orange-brown fall cartoon calendar
A snowy cozy winter cartoon calendar

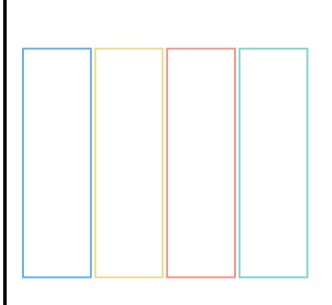 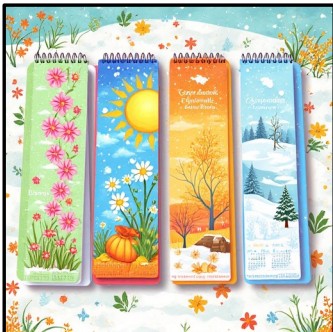

A colorful collection of four cartoon-styled calendars, each uniquely illustrating the essence of a different season. The spring calendar bursts with shades of green and pink, featuring blooming flowers and sprouting leaves. The summer calendar glows with vibrant sun motifs and vivid blue skies. Autumn is represented with warm oranges and browns, showcasing falling leaves and harvest themes. The winter calendar is adorned with soft whites and blues, depicting snowy scenes and cozy fireside images. Each calendar is distinct, yet they all share a whimsical charm that captures the spirit of their respective seasons.

A glazed ceramic plate
A juicy grilled chicken serving
A crispy mix of steamed vegetables
A creamy mashed potatoes scoop
A fresh parsley sprig
A neatly wrapped silverware set
A folded cloth napkin

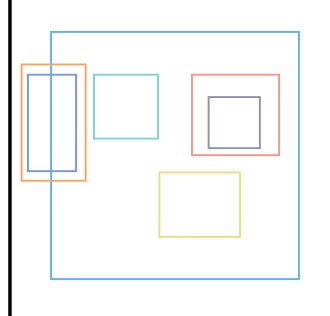 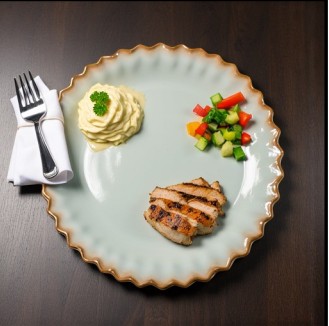

A close-up image of a ceramic plate filled with a colorful assortment of food, including slices of grilled chicken, a mix of steamed vegetables, and a scoop of mashed potatoes garnished with a sprig of parsley. The plate is set on a dark wooden dining table, and beside it lies a set of silverware wrapped neatly in a cloth napkin. The food is arranged in an appetizing display, showcasing a variety of textures from the crisp vegetables to the creamy potatoes.

Layout-free mode : instance decomposition not available

layout-free mode : bounding boxes not available

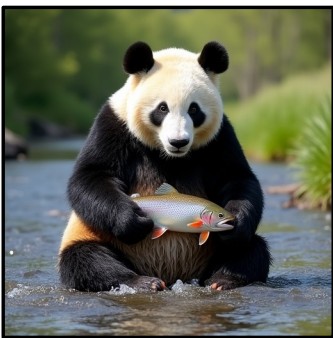

A sizable panda bear is situated in the center of a bubbling stream, its black and white fur contrasting with the lush greenery that lines the water's edge. In its paws, the bear is holding a glistening, silver-colored trout. The water flows around the bear's legs, creating ripples that reflect the sunlight.

Figure 10: Generative results of our IPCP on DPG Bench.

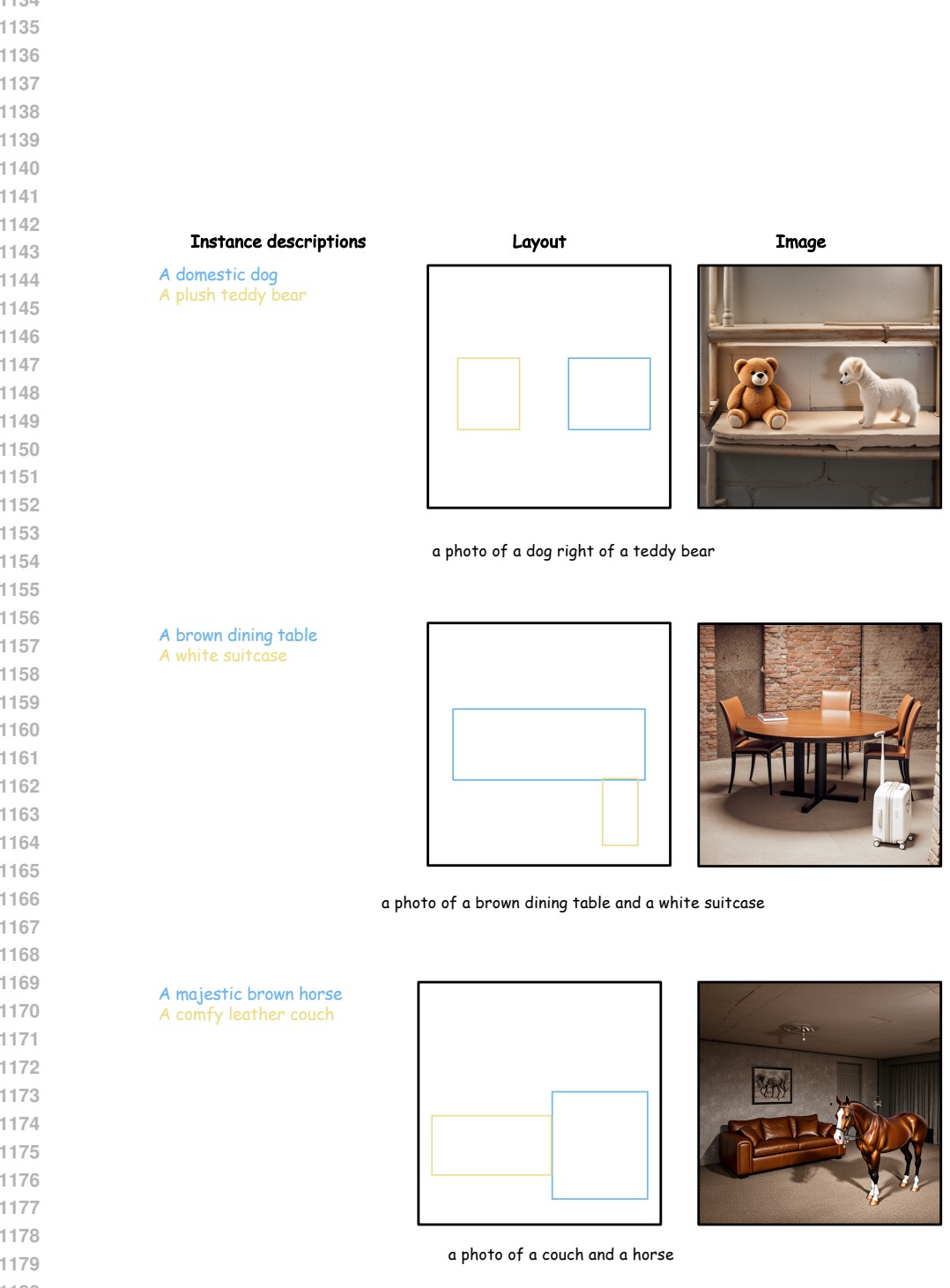

Figure 11: Generative results of our IPCP on Geneval.

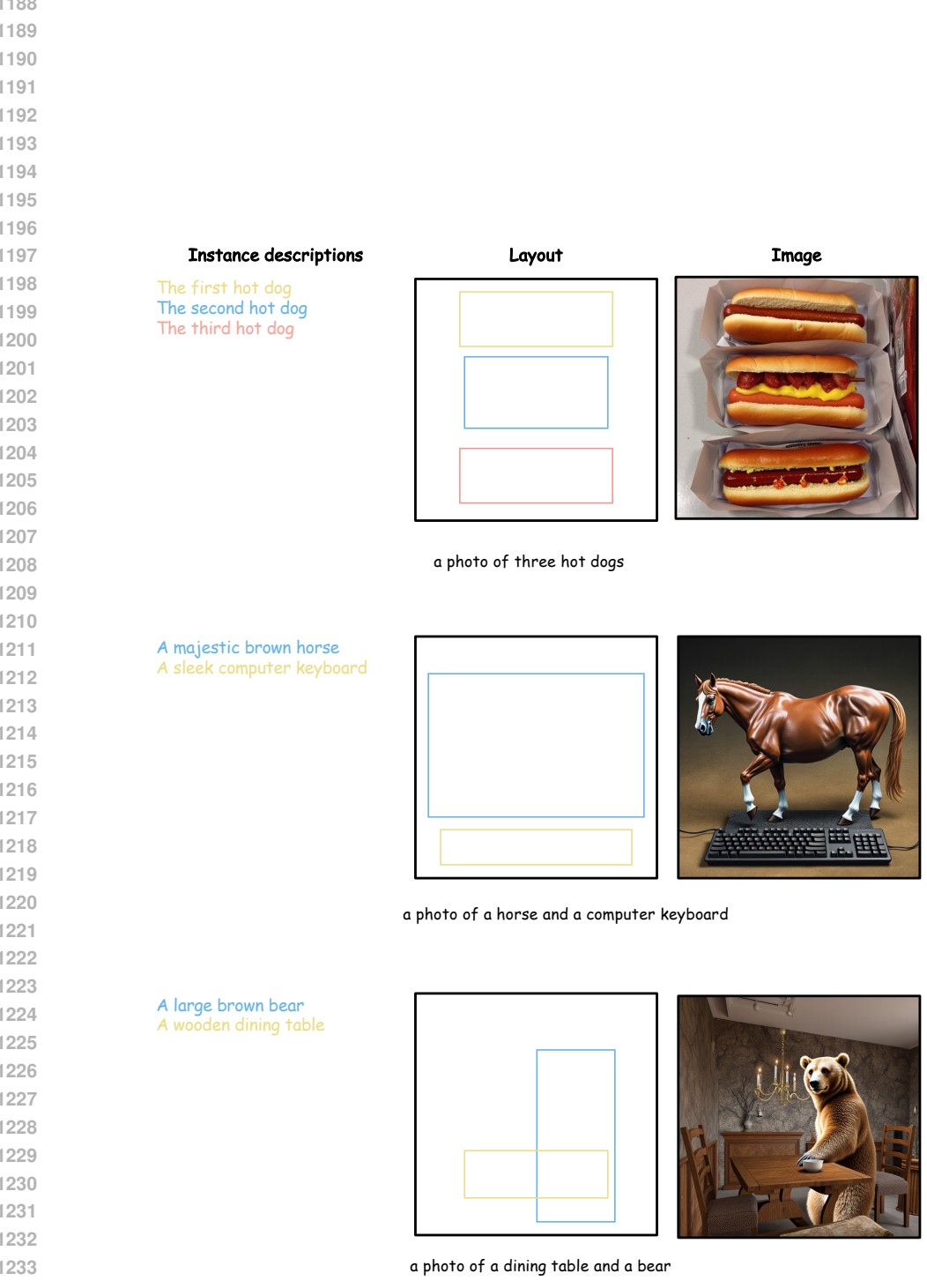

| Instance descriptions | Layout | Image |
| --- | --- | --- |

The first hot dog
The second hot dog
The third hot dog

a photo of three hot dogs

A majestic brown horse
A sleek computer keyboard

a photo of a horse and a computer keyboard

A large brown bear
A wooden dining table

a photo of a dining table and a bear

Figure 12: Generative results of our IPCP on Geneval.

| Instance descriptions | Layout | Image |
|---|---|---|

A stack of blue books
A red apple
A bell
A wooden teacher's desk
A dusty old chalkboard

A dusty old chalkboard fills the background of this image, which features a wooden teacher's desk that has blue books, a red apple, and a bell placed on top.

A man with a relaxed posture
A woman in a bland kitchen
A modern kitchen microwave
A plate with food
An empty kitchen plate
A wine bottle

A man, his arm across the woman next to him, stands in a blandly colored kitchen area, in front of a black-rimmed window, next to a counter with a microwave, plates, with and without food, and wine bottles.

A modern flat screen TV
A large decorative clock
A first leather chair
A second leather chair
A classic fireplace
A bookshelf with many DVDs

A photo of someones living room complete with a bookshelf full of dvds, two leather chairs, a flat screen tv, fireplace, and a overly large decorative clock.

Figure 13: Generative results of our IPCP on COCO.

| Instance descriptions | Layout | Image |
|---|---|---|

A dog with baseball bat
A uniformed baseball player
A uniformed baseball player
A uniformed baseball player
A uniformed baseball player

Four baseball players standing behind a fence in a baseball field while a dog carrying a bat walks across the field.

A wooden vanity
A bathroom mirror
An integrated bathroom sink
A tiled shower
A molded plastic bathtub
A small practical shelf
A small bathroom lamp

a photo of a brown dining table and a white suitcase

A  contemporary sectional sofa
A large decorative mirror
A filled bookshelf
A modern floor lamp
A modern floor lamp
A vase of fresh flowers
A vase of fresh flowers

A sectional sofa in a front room with a bookshelf and mirror with two floor lamps and two vases of flowers on either side。

Figure 14: Generative results of our IPCP on COCO.

A purple-themed bathroom
One bathroom sink
One bathroom sink
One bathroom sink
A purple countertop
A mirror with light bulbs

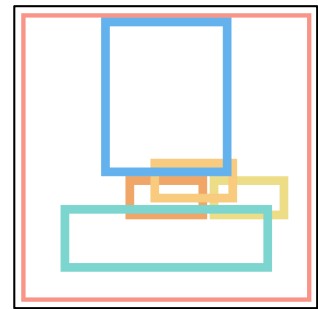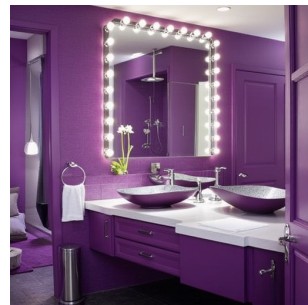

A purple bathroom with three sinks and a purple countertop with a mirror surrounded with light bulbs.

An open airstrip environment
A man in casual attire
A small single-engine airplane
A medium-sized dog
Another medium-sized dog

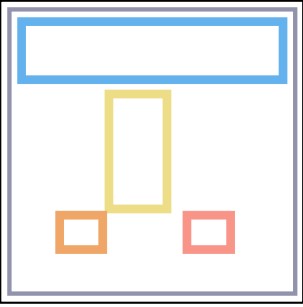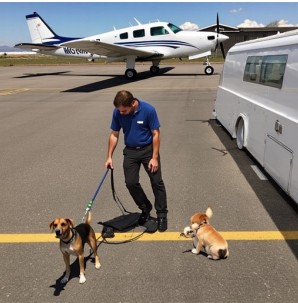

A man standing next to a small airplane with two dogs.

An open field
A standing person observing
A jet taking off
A parked car
A parked car

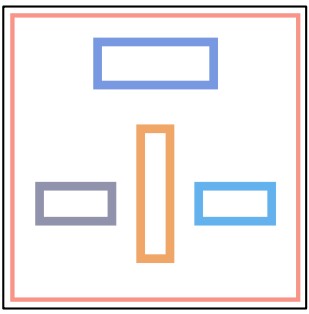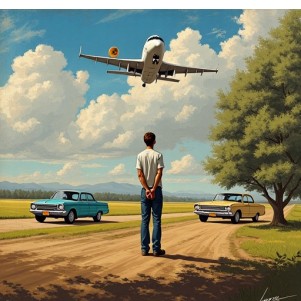

A person standing near two cars is watching a jet take off from a field.

Figure 15: Generative results of our IPCP on COCO.

Ocean with dynamic waves
An active wave rider
A sleek surfboard

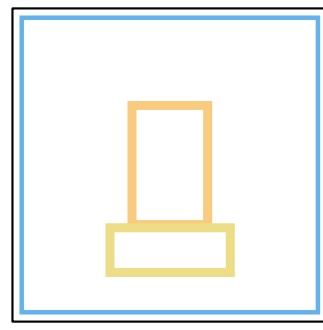 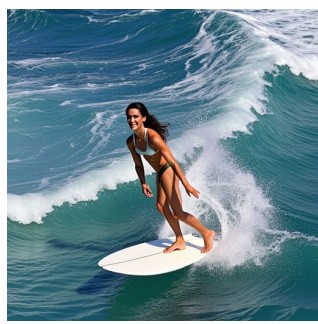

Someone riding waves on their surfboard in the ocean.

Modern kitchen setup
A modern kitchen microwave
A standard kitchen oven
Line of kitchen cabinets

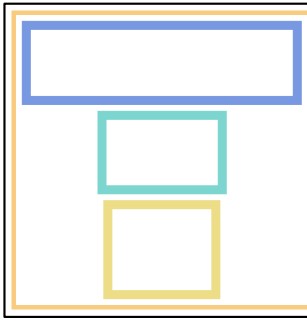 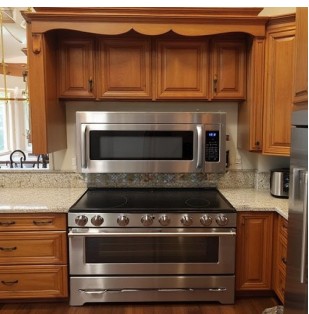

The view of a kitchen's microwave, oven, and cabinets.

A natural fenced area
A black and white zebra
A black and white zebra
A black and white zebra
A tall giraffe
A tall giraffe
A tall giraffe
A tall giraffe

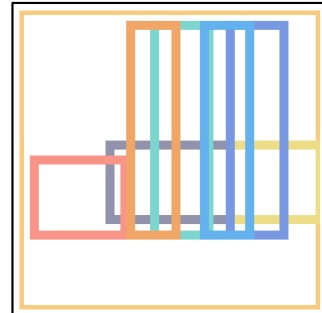 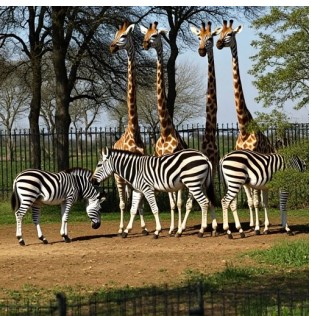

Three zebra and four giraffe inside a fenced area.

Figure 16: Generative results of our IPCP on COCO.

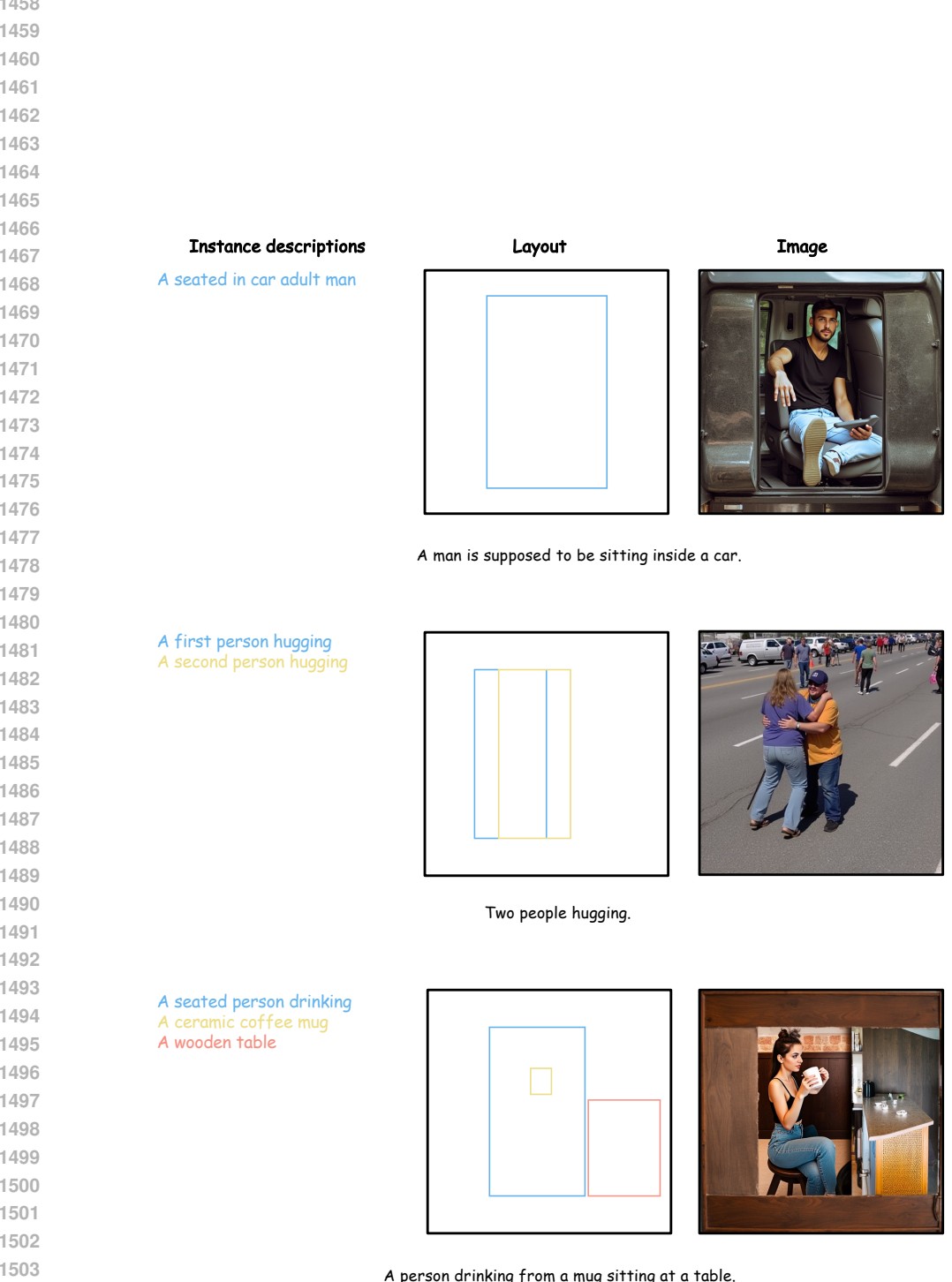

Figure 17: Generative results of our IPCP.

