# OpenReview forum: "IPCP: Interpreter, Planner, Checker, and Painter Dialogue for Compositional Text-to-Image Generation"
_ICLR.cc/2026/Conference — ICLR 2026 Conference Withdrawn Submission_

### Official Review · Reviewer_X5tM · 2025-10-26

**Soundness:** 3
**Presentation:** 3
**Contribution:** 3
**Rating:** 6
**Confidence:** 3

**Summary:**

This paper propose​ a novel multi-agent framework IPCP designed to tackle the challenge of ​​compositional T2I generation​​. IPCP consists of a divide-and-conquer strategy​​, grounding layout decisions in visual context​​, and an ​​explicit error correction process to achieve better T2I in a multi-agent manner.

**Strengths:**

- The paper clearly points out the key limitations of existing methods like quadratic relational complexity, lack of visual grounding, and lack of error correction. The IPCP framework is elegantly designed to address each one directly.
- The roles of the agents designed in IPCP are well-defined and logically complementary.

**Weaknesses:**

- Each generation process requires multiple calls to an MLLM (for the Interpreter, Planner, and Checker) and iterative calls to the Painter. The total latency compared to a single-pass model is likely significantly higher. A discussion on the efficiency and inference time would be valuable for this paper. Is the performance gain balanced with the inference latency?
- The paper uses powerful MLLMs like GPT-5 and Flux but provides limited information on the details of the prompts used in this process, hyperparameters, or the exact architecture of the MLLM agents.
- While qualitative results are strong, a more detailed discussion of the IPCP's limitations and typical failure cases (for exmaple, what happens when the Interpreter misranks object priority? When the Checker fails to correct a major error?) would strengthen the paper.

I am not very familiar with the topic (Text-to-Image) of this paper. Therefore, I hope authors, ACs, SACs, PCs could carefully consider my review comments and reduce its weight in the final decision.

**Questions:**

- For the results in DPG-Bench, why IPCP outperforms baselines like OmniGen2 in the overall performance but underperforms in most of other metrics? I found that in specific metrics, IPCP performs worse than OmniGen2 in 4 out of 5 metrics, yet it significantly outperforms it in overall performance. Is this evaluation method reasonable? Are there any inconsistencies or incoherencies here?

---

### Official Review · Reviewer_n2fr · 2025-10-26

**Soundness:** 3
**Presentation:** 2
**Contribution:** 2
**Rating:** 4
**Confidence:** 4

**Summary:**

The paper presents the IPCP framework, in which four intelligent agents collaborate to generate images: 1) Interpreter, which adaptively selects between direct text-to-image generation and a layout-driven workflow; 2) Planner, which designs the overall layout for the image; 3) Checker, which verifies and refines spatial and attribute consistency; and 4) Painter, which progressively renders each instance onto the final image. This approach demonstrates strong performance on text-to-image benchmarks; however, the paper omits several crucial technical details and contains certain ambiguities, leaving the overall presentation incomplete.

**Strengths:**

The results show that the proposed method performs strongly and is logically coherent.

**Weaknesses:**

The paper omits several crucial technical details and contains certain ambiguities, leaving the overall presentation incomplete. For specific issues, please refer to the Questions section.

**Questions:**

My main concerns are as follows. If these issues are addressed and my doubts clarified, **I would consider raising the score**:

Question 1: The paper states that it employs “3DIS: A 3D-aware diffusion model for instance-level synthesis” for layout-to-image generation. However, I could not find any work with this exact title, which makes it unclear what specific method was actually used. This ambiguity is likely to confuse readers.

Question 2: The collaboration between the Planner and the Painter requires iteratively generating each object. The paper does not explain how this is implemented. Is this achieved using image editing techniques? I believe this is a critical technical detail that should be explicitly described.

Question 3: The proposed method follows an iterative generation paradigm, rendering one object at a time. How does this approach differ from conventional layout-to-image methods that generate multiple objects in a single step? This comparison is important for understanding the contribution.

Question 4: The paper should present examples showing the intermediate results of the full pipeline execution. Could you illustrate how multiple agents collaborate step by step? For instance, how does the Painter progressively render each object, and how does the Checker detect and correct errors?

---

### Official Review · Reviewer_Csij · 2025-10-27

**Soundness:** 3
**Presentation:** 3
**Contribution:** 2
**Rating:** 4
**Confidence:** 4

**Summary:**

This paper presents IPCP, an interactive multi-agent dialogue framework for text-to-image generation, comprising four specialized agents: Interpreter, Planner, Checker, and Painter. The method demonstrates strong performance on GenEval and DPG-Bench for compositional scene generation, multi-object layouts, and attribute alignment. The work is novel, well-motivated, and experimentally thorough.

**Strengths:**

1. The paper has a clear and well-structured organization.

2. The proposed method is reasonable and feasible.

3. The experiments demonstrate that their approach is effective to some extent.

**Weaknesses:**

1.The experiments are somewhat limited, as comparisons are made only with T2I methods.

2.The explanations of each component could be clearer—for example, by providing more visualizations or showing the multi-level generation step by step.

3. Some references in the paper are incorrect.

4. See in questions.

**Questions:**

1. It would be more convincing if the Planner could generate multiple layouts at once and compare them using the L2I model, or if comparisons with existing image-generation agents were provided.

2. Are there examples that demonstrate very complex scenes? Or could the paper show a full end-to-end generation process?

3. When generating layouts at different priority levels, can the Planner observe objects from other levels to arrange positions more reasonably? For example, if the most important object is accidentally placed at the far right, a less important object may not have space to be placed to its right—does this happen, and can the Checker correct such cases?

4. In Table 4, introducing the layout-aware mode causes a drop in the Global metric, and adding Visual context leads to a significant drop in the Other metric—could the authors explain the reasons for this?

Typo: Some references in the paper are incorrect, e.g., 3DIS is cited incorrectly.

---

### Official Review · Reviewer_sWXY · 2025-10-31

**Soundness:** 3
**Presentation:** 3
**Contribution:** 2
**Rating:** 4
**Confidence:** 4

**Summary:**

The paper proposes IPCP, an interactive multi-agent dialogue framework for compositional text-to-image generation.
Four agents—Interpreter, Planner, Checker, and Painter—collaborate in an iterative loop.
Experiments on GenEval and DPG-Bench show large gains over strong baselines such as T2I-Copilot, GoT, and GPT Image, achieving new SOTA performance in compositional fidelity, spatial accuracy, and attribute binding. Ablation studies confirm the importance of visual grounding and the error-correction mechanism.

**Strengths:**

1. The four-agent IPCP pipeline (Interpreter–Planner–Checker–Painter) introduces modular collaboration rather than a fixed pipeline, enabling flexible adaptation between direct and layout-aware modes.
2. The paper clearly articulates three pain points—layout complexity, missing visual grounding, and lack of explicit correction—and ties each to a corresponding agent.
3. The paper is easy to follow.

**Weaknesses:**

1. The authors claim that the proposed method is designed to handle multiple objects and preserve their attributes in complex scenes. However, based on the visualizations provided, the generated examples involve rather simple scenes with only around five objects on average. There is a lack of evaluation on genuinely complex or densely populated scenes that would better test the robustness of the proposed layout-planning mechanism. In such scenarios, would the number of agent calls increase significantly, thereby introducing substantial time and resource costs? A quantitative analysis of scalability under complex settings is needed.

2. Although the automatic metrics (GenEval and DPG-Bench) are comprehensive, a small-scale human evaluation would be valuable to better assess the perceived compositional quality and visual faithfulness of the generated images.

3. Regarding the agent design, the paper introduces four fixed roles (Interpreter, Planner, Checker, and Painter), each with a predefined workflow. However, it remains unclear whether such fine-grained role separation is truly necessary. Could a simpler configuration, such as a creator and a feedback provider, achieve similar effects? Overall, the innovation appears somewhat limited; the design feels more system-driven than principle-driven, without a deep analysis of why an agentic paradigm is inherently suitable for text-to-image generation or how each role contributes indispensably to the overall performance. As it stands, the contribution seems largely engineering-oriented, with modest conceptual novelty.

**Questions:**

1. How sensitive is IPCP to the quality of the underlying T2I/L2I model? For instance, if FLUX is replaced with SDXL, does performance degrade proportionally?

2. Can the Checker handle non-rigid or abstract relations (e.g., “a shadow beneath the cup”)?

3. Did you experiment with different ordering heuristics besides semantic salience for prioritizing objects?

4. How robust is VCoT when the partial canvas contains hallucinated geometry? Is error propagation a problem?

---

### Note · Authors · 2025-11-14

I have read and agree with the venue's withdrawal policy on behalf of myself and my co-authors.